# Adjuvant Vaccination with Allogenic Dendritic Cells Significantly Prolongs Overall Survival in High-Grade Gliomas: Results of a Phase II Trial

**DOI:** 10.3390/cancers15041239

**Published:** 2023-02-15

**Authors:** Guilherme Lepski, Patricia C. Bergami-Santos, Mariana P. Pinho, Nadia E. Chauca-Torres, Gabriela C. M. Evangelista, Sarah F. Teixeira, Elizabeth Flatow, Jaqueline V. de Oliveira, Carla Fogolin, Nataly Peres, Analía Arévalo, Venâncio A. F. Alves, José A. M. Barbuto

**Affiliations:** 1Laboratory of Experimental Surgery (LIM26), Hospital das Clínicas, Medical School, Universidade de São Paulo, São Paulo 01246-903, Brazil; 2Department of Neurosurgery, Eberhard-Karls University, 72076 Tuebingen, Germany; 3Departamento de Imunologia, Instituto de Ciencias Biomedicas, Universidade de São Paulo, São Paulo 05508-000, Brazil; 4Department of Psychiatry, Medical School, Universidade de São Paulo, São Paulo 05403-010, Brazil; 5Department of Pathology, Medical School, Universidade de São Paulo, São Paulo 05403-010, Brazil; 6Laboratory of Medical Investigation in Pathogenesis and Targeted Therapy in Onco-Immuno-Hematology (LIM-31), Department of Hematology, Hospital das Clínicas, Medical School, Universidade de São Paulo, São Paulo 05403-010, Brazil

**Keywords:** glioblastoma, immunotherapy, dendritic cells, clinical trial, cancer vaccine, cancer treatment

## Abstract

**Simple Summary:**

Among all intracranial tumors, 31.5% are malignant, and among those, glioblastomas account for 47%. Recently, our group reported the case of a patient with glioblastoma who underwent vaccination based on dendritic cells and experienced near-complete tumor remission. Here we report the results of a phase I/II prospective, non-controlled clinical trial with 37 patients harboring glioblastoma or grade 4 astrocytomas. Patients received monthly intradermal injections of allogenic dendritic cell vaccinations. The survival curves of the vaccinated populations were compared with patients from the GDC (Genomics Data Commons) database, which revealed that overall survival was 75% greater in the vaccinated glioblastoma group (16 to 28 months, hazard ratio 0.53) and 200% greater in the vaccinated astrocytoma grade 4 group (20 to 60 months, hazard ratio 0.18). Furthermore, seven patients remain alive to this day. We believe that the data reported here can foster the continued improvement of treatment protocols based on cellular immunotherapy.

**Abstract:**

Immunotherapy for cancer treatment has gained increased attention in recent years. Recently, our group reported the case of a patient with glioblastoma who underwent vaccination based on dendritic cells and experienced a strong Th1 immune response together with near-complete tumor remission. Here we report the results of a phase I/II prospective, non-controlled clinical trial with 37 patients harboring glioblastoma or grade 4 astrocytomas. At the time of first recurrence after surgery, patients began receiving monthly intradermal injections of allogenic DC-autologous tumor cell hybridomas. Overall survival, quality of life, and immunological profiles were assessed prospectively. Compared with patients in the Genomic Data Commons data bank, overall survival for vaccinated patients with glioblastoma was 27.6 ± 2.4 months (vs. 16.3 ± 0.7, log-rank *p* < 0.001, hazard ratio 0.53, 95%CI 0.36–0.78, *p* < 0.01), and it was 59.5 ± 15.9 for vaccinated astrocytoma grade 4 patients (vs. 19.8 ± 2.5, log-rank *p* < 0.05, hazard ratio 0.18, 95%CI 0.05–0.62, *p* < 0.01). Furthermore, seven vaccinated patients (two IDH-1-mutated and five wild type) remain alive at the time of this report (overall survival 47.9 months, SD 21.1, range: 25.4–78.6 months since diagnosis; and 34.2 months since recurrence, range: 17.8 to 40.7, SD 21.3). We believe that the data reported here can foster the improvement of treatment protocols for high-grade gliomas based on cellular immunotherapy.

## 1. Materials and Methods

### 1.1. Sample Size Estimation

Sample size was estimated prior to study initiation using JMP, version 16.0.0 (SAS Institute, Cary, NC, USA). To this end, we fitted the survival plot of a historic cohort of 106 patients with glioblastoma from our institution (treated with maximal resection, Temozolomide and fraccioned radiotherapy) with a lognormal curve, which resulted in a scale coefficient of 0.6. Then, we defined the observation period of the present study as 36 months, desired reliability as 80%, and alpha error as 0.05. The calculation resulted in a sample size of 43 patients necessary to test survival at 12 months after vaccine initiation. Indeed, the study was interrupted when a significance level of 0.01 was achieved, after 37 patients.

### 1.2. Patient Recruitment and Ethics

We enrolled 37 adult patients in this phase I/II prospective trial on allogenic DC vaccination for GBM at our institution. Inclusion criteria were >18 years of age, anatomopathological and immunohistochemical confirmation of glioblastoma or grade 4 astrocytoma (according to the WHO 2021 classification) [1], previous treatment according to the best practice (maximal surgical resection, chemotherapy with temozolomide, and fractionated radiotherapy with 60 Gy total dose) [2], Karnofsky performance score 50 or higher, and radiological progression demonstrated by recent MRI, according to RANO criteria [3]. Exclusion criteria were cognitive impairment or aphasia (which would limit comprehension of the consent form), other cancers, pregnancy, other severe or life-threatening clinical conditions, immunodeficiency of any cause, coagulopathy, chronic infection, and incomplete previous treatment according to the best oncological evidence. Patients were enrolled at the time of tumor recurrence. The basic oncologic treatment during the vaccination trial varied among patients, as each patient followed the recommendations of the interdisciplinary tumor board in charge of their treatment. As the present study was a non-controlled phase II trial, the scientific committee did not interfere with clinicians’ choice for better concomitant chemotherapy, when recommended. All procedures were approved by the institutional Ethics Committee and the National Research Council at the University of São Paulo (approval No. 58882116.7.3001.0065), and patients were enrolled after providing written informed consent. Clinical and laboratory data were collected prospectively, anonymously, and recorded using the RedCap platform hosted at Hospital das Clínicas, Medical School, University of São Paulo (https://redcap.hc.fm.usp.br, accessed on 5 December 2022) [4,5].

### 1.3. Sample Tissue Collection and Processing

The fresh tumor sample obtained from surgical resection was minced and digested with collagenase type VIII (0.56 mg/mL; Sigma-Aldrich, San Luis, MI, USA), under agitation, at 37 °C for 2 h. Cell suspensions were separated from the non-digested fragments using sterile gauze and washed twice in RPMI-1640. In parallel, a small portion of the tumor was fixed in paraformaldehyde 4% and processed for pathological analysis (hematoxylin/eosin and immunohistochemistry). IDH-1 status was determined by immunohistochemistry (monoclonal antibody clone H09 specific for the mutation R132H).

### 1.4. Vaccine Production and Application

Peripheral blood mononuclear cells (PBMCs) were obtained from leukapheresis chambers of blood donors by separation over Ficoll-Paque gradient (GE Healthcare). PBMCs (3 × 10^8^) were seeded in 75 cm^2^ flasks and incubated for 2 h at 37 °C and 5% CO_2_. After incubation, nonadherent cells were removed, and adherent cells were cultured in AIM-V supplemented with GM-CSF (50 ng/mL; Peprotech, Cranbury, NJ, USA) and IL-4 (50 ng/mL; Peprotech). After five days, the cells received a maturation stimulus with TNF-α (50 ng/mL; Peprotech), and 48 h after activation, were harvested and resuspended in a sterile 5% glucose solution; tumor cells were thawed, washed, and also resuspended in a sterile 5% glucose solution. Both cell suspensions were at a concentration of 1 × 10^7^ cells/mL. The two cell suspensions were mixed, and the cells were fused by an electric pulse of 1000 V/cm at 25 μF (applied by a Gene-Pulser II; Bio-Rad, Richmond, CA, USA), after being aligned in an electrical field (62.5 V/cm) for 15 s. Cells were left to rest for 2 min in the electroporation cuvette and transferred to a relaxation buffer (100-mM KCL, 3-mM NaCl, 1.25-mM EDTA, 10-mM PIPES, 0.5-mM ATP, adjusted to pH 6.8), where they were kept for an additional 3 min. The hybrid cell preparation was centrifuged, resuspended in 1 mL of sterile phosphate-buffered saline (pH 7.2), and after irradiation (200 Gy), injected into each patient [6]. The harvested tumor samples were sufficient to produce 1 to 12 vaccine doses (mean 5; median 4). Freshly prepared hybrid cell suspensions were applied once a month intradermically, in 2 points in the forearm, 0.5 mL each, after proper asepsis with alcohol swabs. For patients who were receiving temozolomide as adjuvant therapy, the vaccine was applied on the 14th day of the cycle, or whenever the leukocyte counts returned to normal.

### 1.5. Follow-Up

All study patients were followed up at monthly intervals. Neurologic status was assessed by general neurologic exam and the Mini-mental status exam. Overall performance was assessed by the Karnofsky performance status (KPS) and WHO-ECOG, whereas global health and quality of life were assessed by the EORTC-QLQ-C30, EORTC QLQ-BN20, FACT-Br, and MDASI-BT evaluation scales. Magnetic resonance imaging (MRI) scans where scheduled every 2 months, and tumor progression was defined according to RANO criteria [3]. Adverse events were categorized according to the US National Cancer Institute’s Common Terminology Criteria for adverse events version 4.0 [7]. For statistical comparisons on survival estimates for high grade gliomas (both with or without R132H IDH-1 mutation), we accessed the open data available on the Genomic Data Commons (GDC) [8] database, National Cancer Institute (NCI, USA), National Institutes of Health (NIH, USA), which is mostly based on the open source data from the Cancer Genome Atlas (TCGA) Research Network [9].

## 2. Results

The current DC-vaccination trial included 37 patients with a diagnosis of recurrent glioblastoma (IDH-1 wild type, n = 28, 76%) or grade 4 astrocytoma (IDH-1 mutated, n = 9, 24%). Mean age was 47 (SD 13, ranging from 19 to 75), and 14 patients (38%) were female. During vaccination, 10 patients (27%) were receiving temozolomide, 9 (24%) bevacizumab, 7 (19%) lomustine, and 6 (16%) were receiving various combinations of the drugs mentioned above. Disease duration prior to study enrollment was 14.8 ± 11.0 months for the patients with glioblastoma and 45.5 ± 20.3 months for the patients with grade 4 astrocytoma. Noteworthy, these periods do not refer to time to first recurrence, given that most patients had experienced more than 1 recurrence prior to enrollment. The study population received 1 to 12 vaccine doses (mean 5, median 4). Vaccination was interrupted by death or after all frozen tumor cell samples (maximum 12) had been applied. 

Figure 1 illustrates the study phases and treatment course in a 28-year-old male patient with glioblastoma (IDH-1 wild type). For this patient, gross total resection was not possible because of the tumor’s proximity to the motor projection pathways. After chemoradiotherapy (60 Gy concomitant with temozolomide), the tumor recurred, and the patient was submitted to a second-look resection followed by seven doses of the DC vaccine. The MRI sequences show an almost-complete remission of the Flair hyperintensity, which was accompanied by a very satisfactory clinical evolution. More specifically, the patient progressively recovered from a grade 4 motor weakness on the right side, and resumed his normal life, going back to work and taking care of himself. Temozolomide was interrupted 11 months after vaccine initiation, and the patient is alive today, completely asymptomatic, 24 months after trial initiation.

Among all 37 vaccinated patients harboring high-grade gliomas (considered to be glioblastomas before the new WHO classification of CNS tumors from 2021), overall survival (OS) was 26.9 months (95% CI 22.3 to 33.2), which we compared with matched patients from the Genomic Data Commons (GDC) data bank (n = 595, mean OS 12.1, 95% CI 10.9 to 12.8, hazard ratio 0.45; 0.32–0.64; *p* < 0.0001). Considering time from relapse, survival probability was 16.6 months (95% CI 1.5–54.8). For specific time points from relapse and from diagnosis, survival was 61.3% at 6 months from relapse (24 months from diagnosis), 46.6% at 12 (30) months, 34.9% at 18 (36) months, 26.0% at 24 (42) months, and 19.3% at 30 (48) months.

To assess the specific effect of the vaccine on high grade glioma subgroups according to the newest brain tumor classification, we compared IDH-1 mutated cases form our series (today considered to be grade 4 astrocytomas, n = 9) with 23 patients in the GDC data bank; we also compared the patients in our group who had glioblastoma (n = 28) with 572 patients in the GDC with wild type IDH-1 (today considered glioblastoma). Both vaccinated groups had significantly longer survival times than GDC controls: in the astrocytoma 4 comparison, vaccinated patients had a mean survival of 59.5 ± 15.9 months, and the GDC group had a mean survival of 19.8 ± 2.5 months (log rank *p* < 0.01, HR= 0.18, 95%CI 0.05–0.62, *p* < 0.01). This indicates an 82% relative reduction in risk of death at any time point for patients with grade 4 astrocytoma. For glioblastoma, on the other hand, survival was 27.6 ± 2.4 months in the vaccinated population, versus 16.3 ± 0.7 months in the GDC group (log-rank *p* < 0.001, HR = 0.53, 95%CI 0.36–0.78, *p* < 0.01; see Figure 2), which indicates a 47% reduction in risk of death at any time point after vaccination for patients with glioblastoma. When we calculated survival since the start of the vaccination program, patients harboring astrocytoma 4 lived for 13.1 ± 16.1 months, whereas patients with glioblastoma survived for 12.4 ± 10.9 months. Moreover, survival was not correlated with number of vaccine doses (Spearman’s rank, *p* > 0.05).

Most impressively, 7 of the 37 patients enrolled in the present trial remain alive today (two with mutated IDH-1 and 5 with wild type). Overall survival of those seven patients was 47.9 ± 21.1 months, and survival after the start of vaccination was 34.2 ± 21.3 months. For glioblastoma, the overall survival range in this small subgroup of long responders (n = 5) was between 25.4 and 45.3 months; similarly, overall survival was 76.5 and 78.6 months in the two patients with astrocytoma 4. In terms of survival time after vaccination, the range was 17.8 to 36.7 for glioblastoma and 35.5 and 40.7 for grade 4 astrocytoma.

Regarding safety, we observed just one case of mild and transitory hepatitis, which may have been associated with vaccination, but might also be explained by concomitant Lomustine treatment. The patient developed nausea and epigastric pain after the sixth vaccination dose, and their liver enzymes increased transitorily. No alteration in the vaccination regimen was made, but Lomustine was paused. After that, symptoms subsided, and the enzymes gradually returned to normal. A second patient developed hepatitis during vaccination, but in that case, there was a clear etiological association with acute hepatitis A infection, demonstrated by IgM levels and specific antibodies. Thus, the relationship with vaccination in this case is improbable. The symptomatology was light and transitory, and the vaccination schedule remained unaltered. No other light, moderate, or severe adverse events were observed.

Next, we plotted the lymphocyte over neutrophil counts throughout the vaccination period (Figure 3) and observed a bi-phasic pattern of behavior, with a narrow peak at the second month of vaccination and a broader peak between six and eight months, followed by a consistent decline in the Ly/Neutro ratio.

## 3. Discussion

Glioblastoma is the most frequent malignant primary brain neoplasia in adults [10]. Despite recent advances in imaging technology, surgical procedures, and adjuvant therapies, glioblastoma remains highly resistant to treatment [11,12]. Its poor prognosis has been explained by high intra-tumor heterogeneity, very fast doubling-time, a highly infiltrative nature, and strong induction of neo-vascularization [13], features to which another key point must be added: its immunosuppressive microenvironment [14]. These characteristics make glioblastoma a very dynamic and constantly changing cancer, with high adaptability to its microenvironment, and consequently, high resistance to therapeutic attempts. 

The estimated overall survival for glioblastoma is 14.6 months overall [2]—1.1 years for IDH-1 wild type and 3.6 years for IDH-1 mutant subgroups [15], presently classified as grade 4 astrocytomas [1]. Disease progression seems virtually inevitable and occurs at a median of 6.9 months [2]. Clinical decisions are usually made on an individual basis, and no consensus exists on what those decisions should be [16]. Glioblastoma recurrence is currently treated with a number of different strategies, and all of them face countless challenges [17]. Immunotherapy-based treatments have recently gained increased attention, and while various approaches are currently under investigation, few initial promising results have been confirmed in larger studies [18].

One immunology-based strategy (antigen-unspecific) involves immune checkpoint inhibitors [19]. In this method, antibodies target proteins involved in the inhibition of effector T-cells and activation of regulatory T-cells. Indeed, anti-programmed cell death protein 1 (PD-1) and anti-programmed cell death ligand 1 (PD-L1) received FDA approval for melanoma in 2014. While the efficacy of the anti-PD-1 agent nivolumab [20] and the anti-cytotoxic T-lymphocyte-associated protein 4 (CTLA-4) agent ipilimumab [21] have been recently reported for glioblastoma, there is currently no evidence of these drugs’ superiority over bevacizumab for recurrent glioblastoma. However, in a recent observation (n = 35), the neoadjuvant use of the anti-PD-1 agent pembrolizumab was shown to upregulate T-cell and interferon-γ and improve overall survival (hazard ratio 0.39 for neoadjuvant/adjuvant, *p* = 0.04) [22].

Using a novel approach based on a “library” of unmutated HLA-processed antigens identified from 30 glioblastoma patients, Hilf et al. investigated the efficacy of a personalized, antigen-specific, multi-peptide vaccine in the GAPVAC-101 phase I trial [23]. The vaccine induced strong and sustained responses of CD8^+^T-cells. Contrary to native proteins, tumor-specific antigens are exclusive to tumor cells, thereby more frequently eliciting robust immune responses. However, among the 15 patients enrolled in the above-mentioned trial, 11 presented class 2 adverse events (mainly bone marrow suppression), two patients experienced anaphylactic shock, and one suffered from severe brain edema.

Another promising strategy involves CAR T cells [24] (cantigen receptor), which are genetically modified to target specific tumor-associated antigens independently of major histocompatibility antigen presentation. The major limitation of the CAR T approach against glioblastoma is the scarcity of tumor-specific antigens that are homogeneously and widely expressed. Some candidates are EGFRvIII [25], IL-13Rα2, and HER2. Following this path, Brown et al. reported complete tumor regression of lesions in the brain and spine in one patient treated with intra-ventricular infusion of IL-13Rα2-targeted CAR T cells [26]. In another study where 17 patients were treated with HER2-targeted CAR T, Ahmed et al. observed a partial response in one patient and disease stabilization for 24 months in another three [27]. It is important to note that the marked heterogeneity of glioblastomas, their low mutational load, and dynamic antigenic expression may render CAR T-based therapies impractical for this disease.

Differently from the strategies reported above, vaccines display some attractive features. In contrast to approaches where either a direct effector is used (like CAR-T cells), or checkpoint inhibitors (which release an already established response), vaccines aim to induce active immune responses. The plasticity of such responses could potentially track down and control resistant cell clones within the tumor bulk that inevitably arise in response to therapy. This, however, would only occur in vaccines that do not focus on a single antigen but instead use a set of tumor antigens like those found in tumor lysates, which are efficiently processed and presented by DCs. As the main antigen-presenting cells, DCs seem to be unique in their ability to change a tolerogenic into an immune-responsive state [28,29]—a game-changer in cancer, where the immune balance has already been set towards tolerance.

In a recent publication of our first case with recurrent glioblastoma treated with allogeneic DC vaccination [30], we highlighted the decisive role of CD4^+^T-cells, and to a lesser extent CD8^+^T-cells, during the responsive phase, and treatment failure due to a shift in the immune response to a Th17 response pattern. Despite late recurrence, the patient survived 63 months, significantly longer than expected for similar cases. Here we report on the overall survival of the whole patient series, which includes 37 patients with high grade gliomas—28 diagnosed with glioblastoma and nine with grade 4 astrocytoma. The survival curves of the vaccinated populations were compared with patients from the GDC (Genomics Data Commons) database. These comparisons revealed that overall survival was 75% greater in the vaccinated GBM group (16 to 28 months, log rank *p* < 0.001) and 200% greater in the vaccinated astrocytoma 4 group (from 20 to 60 months, *p* < 0.05).

The strategy reported in our study differed from most previous reports in that we used allogeneic monocyte-derived DCs. We did this to circumvent a frequently observed bias of cancer patients’ monocyte-derived DCs that favors the induction of regulatory T cells [31]. This approach, however, could add a complicating factor to vaccine preparation: the need to use HLA-compatible monocyte donors. To bypass this obstacle, we fused tumor cells with dendritic cells by electroporation, thereby creating heterokaryons, which are effective antigen-presenting cells [32] and confer another valuable feature to the vaccine: the immune-enhancing allogeneic effect [33,34]. This modification in vaccine production might explain why our results differ from those of the Austrian trial Audencel [35] or the Californian phase I/II trial [36], which failed to report any significant changes in overall survival of patients treated with autologous DCs pulsed with tumor lysates. Although they observed no differences in survival, the Austrian group later reported an upregulation of the Th1 immune response, which we also reported in our previous work on the first case of this trial [37]. In another attempt based on autologous DC’s [38], these were pre-pulsed with six different synthetic peptide epitopes targeting tumor or cancer stem cells. Nevertheless, the authors failed to demonstrate any significant difference in overall survival induced by vaccination (2 months, beyond statistical significance). Following this path, another recent phase I trial [39] reported initial observations on 11 patients treated with autologous DCs pulsed with lysate derived from a GBM stem-like stem cell line.

In a multicentric prospective trial published in 2018 including 94 sites in 4 countries [40], 331 patients were enrolled to receive vaccination with autologous DCs pulsed with tumor lysates (DCVax). The whole intended-to-treat population showed survival rates of 89.3%, 46.2%, and 25.4% after 1, 2, and 3 years, respectively. In our series, these probabilities were 92.5%, 61.3%, and 34.9%, respectively. Noteworthily, Liau et al.’s interim survival analysis included the whole ITT population, among which 86.4% had been vaccinated at that time point. Importantly, this study supports the feasibility and safety of adding DC vaccination to the standard first-line therapy against glioblastoma and provided the first evidence that it may extend overall survival.

More recently, that group published the final analysis of that same study [41]. For 232 newly diagnosed glioblastomas, mean OS from surgery was 22.4 months for vaccinated patients. For 64 recurrent glioblastomas, mean OS from relapse was 13.2 months (95%CI, 9.7–16.8).

There are significant differences between Liau et al.’s 2023 report and ours which prevent a direct comparison of results. First, their strategy was based on the use of autologous DCs pulsed with tumor lysates, and we used allogenic DCs electrofused with living autologous tumor cells; second, in Liau et al.’s study, patients were enrolled during first-line treatment, just after surgery and radiotherapy, and 64 patients crossed over to DC vaccination after recurrence (as opposed to our study, which included only recurrent cases); third, Liau et al.’s patients were stratified according to MGMT status, and no IDH-status was presented. Since the most recent WHO classification from 2021, IDH-1 status gained relative importance over MGMT, and cases previously considered to be glioblastoma are now categorized as either glioblastoma (IDH-1 wild type) or astrocytoma grade 4 (IDH-1 mutated), two groups with very different survival rates. In our study, we stratified patients according to IDH-1 status and presented the survival curves from both subgroups separately. Due to these differences, we decided to directly compare the recurrent cases in Liau et al.’s report with all high grade gliomas in our study: while their group had a survival of 13.2 months from relapse (CI95% 9.7 to 16.8), with a hazard ratio of 0.58 (0.00 to 0.76, *p* < 0.01), survival in our series was 16.6 months after relapse (95%CI 1.5 to 54.8, hazard ratio 0.45, 95%CI 0.32 to 0.64, *p* < 0.0001). These different results may be explained by demographic differences between study populations, different proportions of IDH-1 mutated cases in the two series, and differences between vaccines (allogenic vs. autologous DCs). Regardless of these differences, the low significant hazard ratios reported in the two prospective trials highlight the positive effect of DC vaccination in the natural course of high-grade gliomas. This effect may be even greater than that observed when temozolomide was first introduced in the standard care for this disease.

Of critical interest is the long survival observed in 7 of the 37 vaccinated patients (2 with mutated IDH-1 and 5 wild type), who remain alive to this day (mean OS of 48 months, or 34 months after vaccination began) without evidence of tumor recurrence. To further our understanding of our patients’ disease and recovery process, we are currently analyzing immune cell infiltrates within the tumors at the time of surgical resection and changes in circulating lymphocytes throughout the vaccination period. Initial observations suggest that PD1+-lymphocyte counts [42], or low B7-H4 expression levels [43], for instance, might represent a good prognostic factor for patients treated with DCs. These analyses should bring important insights about the ability of DCs to enhance specific anti-tumor T cells, and thus to induce tumor control. 

## 4. Conclusions

Here we report the results of a phase I/II prospective, non-controlled clinical trial with 37 patients harboring glioblastoma or grade 4 astrocytomas who received monthly intradermal injections of allogenic dendritic cell vaccinations. Compared with patients from the GDC database, overall survival was 75% greater in the vaccinated glioblastoma group (HR 0.18, i.e., 82% relative reduction in risk of death at any time point) and 200% greater in the vaccinated astrocytoma 4 group (HR 0.53, i.e., 47% reduction in the risk of death). Furthermore, seven patients remain alive to this day.

Thus, the findings reported in the present study are an important contribution to the field of cellular immunotherapy against cancer, and specifically high-grade gliomas. We hope these encouraging findings guide us and our colleagues in the field in our search for novel strategies against one of the most challenging cancer variations in humans, thereby providing renewed hope for patients and their families.

## Figures and Tables

**Figure 1 cancers-15-01239-f001:**
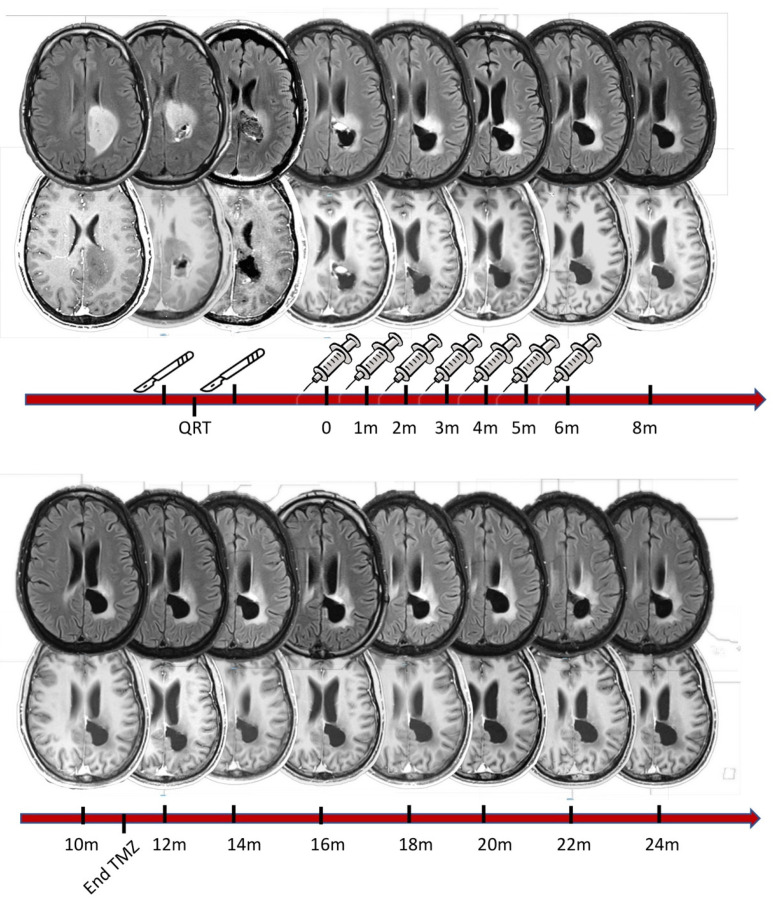
Illustrative case of a 28-year-old male patient whose initial symptoms were motor weakness and partial motor seizures in the right leg. The first column of MRI scans (**top panel**, left) represents the patient’s first MRI scan (**top**: flair sequence; **bottom**: T1 with gadolinium). The images seen in the second column were acquired following a first surgical resection in which roughly 50% of the tumor mass was removed. The first pathology exam revealed glioblastoma, IDH-1 wild type (according to the WHO classification from 2021). The patient then began chemotherapy with temozolomide, concomitant with 60 Gy radiotherapy fractioned into 30 sessions. Three months after the first surgery, the tumor had progressed, which led to a second surgical resection (indicated by scalpels in the figure). This resection was limited anteriorly when electrophysiological monitoring detected deteriorating motor evoked potentials on the right side of the body. The post-operative MRI showed remaining tumor tissue. At that time, the patient was already enrolled in the trial and received the first vaccination dose 3 months after the second surgery. In total, he received 7 doses at monthly intervals, as illustrated above, concomitant with 12 maintenance cycles of temozolomide, which finished 11 months after vaccination had begun (see bottom timeline). The patient underwent an MRI scan every two months to monitor tumor growth. The tumor remained stable throughout the vaccination period and thereafter. In functional terms, the patient is performing 100% according to the Karnofsky performance scale (completely asymptomatic) and 0 (zero) on the Ecog performance score. He has a mini-mental status score of 30, without any impairment in his daily living activities. He regularly takes two anticonvulsant agents, which are enough to control his epileptic seizures. To date, regular MRI scans have shown no tumor progression. He has not received any other chemotherapy treatment since discontinuation of temozolomide one year ago at the time of this report.

**Figure 2 cancers-15-01239-f002:**
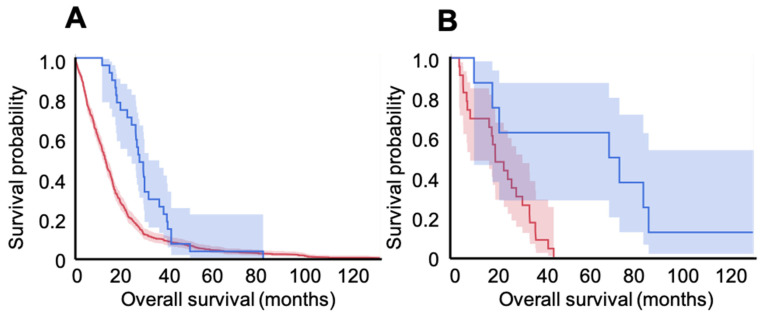
Kaplan–Meier survival curves and respective 95% confidence intervals for vaccinated (blue) versus unvaccinated GDC bank (red) patients with glioblastoma (**A**) and those with grade 4 astrocytoma (**B**). A. In the GBM group, OS was 16.3 ± 0.7 months for the GDC patients (n = 572) and 27.6 ± 2.4 months in the vaccinated population (n = 28; log-rank *p* < 0.001). In the Astro-4 group, OS was 19.8 ± 2.5 months for the GDC population (n = 23) and 59.5 ± 15.9 for the vaccinated patients (n = 9; log-rank *p* < 0.01).

**Figure 3 cancers-15-01239-f003:**
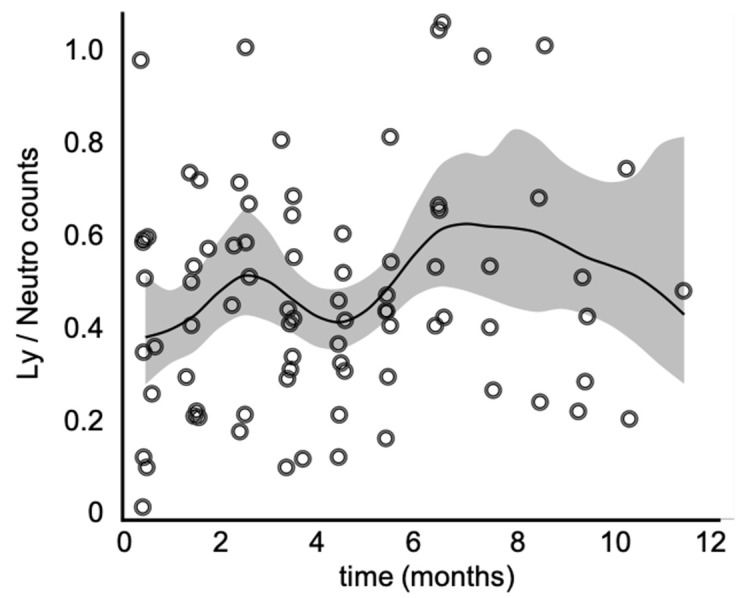
Lymphocyte over neutrophil counts throughout the vaccination period. The continuous line represents the smoothing spline function, and the shaded interval corresponds to the 95% confidence interval. Note the biphasic pattern of behavior, where the first peak is a narrow one, observed by the 2nd month of vaccination, and the second, a more broad-based one, between the 6th and 8th months.

## Data Availability

Raw data and additional information regarding the study patients are available upon request.

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
