# Peer review of "Adjuvant Vaccination with Allogenic Dendritic Cells Significantly Prolongs Overall Survival in High-Grade Gliomas: Results of a Phase II Trial"

_cancers, 2023, doi:10.3390/cancers15041239_

Round 1
Reviewer 1 Report
Immunotherapy for cancer treatment has gained increased attention in recent years. This paper the authors report the results of a phase I/II prospective, non-controlled clinical 34 trial with 37 patients harboring glioblastoma or grade 4 astrocytomas. It’s a very important issue in brain tumor research.
Most impressively, seven of the 37 patients enrolled in the present trial remain alive 175 today (2 with mutated IDH-1 and 5 with wild type). Overall survival of those seven patients is 47.9 ± 21.1 months, and survival after the start of vaccination is 34.2 ± 21.3 months. 177 For glioblastoma, the overall survival range in this small subgroup of long responders 178 (n=5) was between 25.4 and 45.3 months; similarly, overall survival was 76.5 and 78.6 in 179 the two patients with astrocytoma 4. In terms of survival time after vaccination, the range 180 was 17.8 to 36.7 for glioblastoma and 35.5 and 40.7 for grade 4 astrocytoma.
The results of this research are very impressive, but the statistical methods should be taken care because it will determine the clinical trial is successful or not. Instability of Sample Size Calculation and other statistical problems will be mentioned, if possible.
Author Response
We thank the reviewer for his/her very relevant and useful comments. We have added information about sample size estimation to the beginning of Materials and Methods (please see revised manuscript). Additionally, in order to provide a more precise picture of the results obtained, we now show survival curves and the respective pointwise 95% confidence intervals, because of their dependence on sample size. Moreover, in order to allow direct comparisons of our results with those of other clinical trials published on this matter, we included hazard ratios and time-point estimations.
Reviewer 2 Report
In this manuscript, Barbuto and co-workers the results of a phase I/II prospective, non-controlled clinical trial with 37 patients with high grade gliomas which shows that immunotherapy increases overall survival dramatically over the unvaccinated group. Results presented while preliminary are promising and of course, a larger, phase 3, controlled trial is required to confirm the findings. The clinical report is well presented and can be accepted with minor modifications which includes the comparison of their results to those of the other scientific studies that have evaluated the use of immunotherapy for high grade gliomas in the discussion section. Considering this is a phase I/II study, the authors should include any and all a cute and chronic side-effects that were experienced by the patients that were enrolled in this study.
Author Response
We thank the reviewer for his/her excellent comments. In our original submission, we discussed other immunotherapy-based strategies in the discussion, including checkpoint inhibitors, multi-peptide vaccines, and CarT, In the revised version, we have added much more detail about trials based on DCs, like our own. Importantly, we added a report on a multicenter phase III trial for glioblastoma using autologous DCs that was published last month (January 2023). To take it one step further, we revised our survival analysis to allow for direct comparison of our results with those of the cited study. Interestingly, the direct comparison showed our results to be slightly superior. We now discuss the potential explanation for these differences, including the possibility that allogenic cells provide a more efficient immune response.
Regarding safety, we have also added the only two light adverse events observed in this series, and discuss their causality.